# Application of Entropy Spectral Method for Streamflow Forecasting in Northwest China

**DOI:** 10.3390/e21020132

**Published:** 2019-02-01

**Authors:** Gengxi Zhang, Zhenghong Zhou, Xiaoling Su, Olusola O. Ayantobo

**Affiliations:** 1College of Water Resources and Architectural Engineering, Northwest A&F University, Yangling 712100, China; 2Key Laboratory of Agricultural Soil and Water Engineering in Arid and Semiarid Areas, Ministry of Education, Northwest A&F University, Yangling 712100, China; 3Department of Water Resources Management and Agricultural-Meteorology, Federal University of Agriculture, PMB 2240, Abeokuta 110282, Nigeria

**Keywords:** burg entropy, configurational entropy, relative entropy, spectral analysis, streamflow forecasting

## Abstract

Streamflow forecasting is vital for reservoir operation, flood control, power generation, river ecological restoration, irrigation and navigation. Although monthly streamflow time series are statistic, they also exhibit seasonal and periodic patterns. Using maximum Burg entropy, maximum configurational entropy and minimum relative entropy, the forecasting models for monthly streamflow series were constructed for five hydrological stations in northwest China. The evaluation criteria of average relative error (*RE*), root mean square error (*RMSE*), correlation coefficient (*R*) and determination coefficient (*DC*) were selected as performance metrics. Results indicated that the RESA model had the highest forecasting accuracy, followed by the CESA model. However, the BESA model had the highest forecasting accuracy in a low-flow period, and the prediction accuracies of RESA and CESA models in the flood season were relatively higher. In future research, these entropy spectral analysis methods can further be applied to other rivers to verify the applicability in the forecasting of monthly streamflow in China.

## 1. Introduction

Accurate streamflow forecasting is vital for flood control, reservoir management, restoration of river environment, irrigation, and navigation, among other uses [1]. Moreover, it can also provide guidelines for policy makers in the utilization and management of water resources and the formulation of water environmental health protection plans. So far, the simulation of monthly streamflow is a hotspot for hydrologic researchers but is still in exploration and development due to the limitations of forecasting methods. As a traditional method, time series analyses such as autoregressive (AR) or autoregressive moving average (ARMA) models are often used to simulate streamflow, but they cannot address the issue of seasonality that exists in the monthly streamflow series [2]. Fortunately, entropy spectral analysis can extract significant information from streamflow process and forecast monthly streamflow accurately coupled with the time series analysis method. Actually, the spectral method has been successfully used by some researchers for monthly streamflow forecasting with different types of entropy including Burg entropy [3], configuration entropy [1,2], and minimum relative entropy [4,5].

Burg [6] proposed Burg entropy theory (BET) in the frequency domain and then further developed the maximum Burg entropy spectral method (BESA) for time series forecasting. As a classic method for hydrologic forecasting, BESA has been widely used in groundwater level forecasting [7], flood forecasting [8], and streamflow forecasting [3] and has shown an advantage for long-term streamflow forecasting. However, BESA has lower resolution in determining multi-peak spectra, and the monthly streamflow hardly exist in only one period. Maximum configuration entropy spectral method (CESA) is a substitute for the forecasting of multi-peak spectra series.

The concept of the maximum configuration entropy spectral method (CESA) was initially proposed by Frieden [9] in the identification of images. Thereafter, Gull and Daniell [10] applied the concept in the field of astronomy for image reconstruction. In the field of time series analysis, the CESA performs better than the BESA in the determination of spectral density function in the ARMA model and the MA model but has no practical advantage in the AR model [11]. The CESA has been applied for streamflow forecasting by Cui [2] and has shown better reliability than BESA.

As an extension of BESA, minimum relative entropy spectral analysis (RESA) proposed by Shore [12,13] was also applied to the time series forecasting. In RESA, the spectral power was considered as a random variable. Tzannes et al. [14] and Woodbury and Ulrych [15] developed RESA and extended the theory and practice of minimum relative entropy. The RESA spectra have higher resolution and are more accurate in detecting peak location than other methods for spectral computation [16]. The RESA method has been used for monthly streamflow forecasting [4,5,16] and has smaller errors than the other two entropy spectral methods.

However, there is very little research that has reported the application of these methods in streamflow forecasting in China. Moreover, not many researchers have given attention to the selection of streamflow length for a training period. Therefore, the main objectives of this paper are (1) to use three entropy spectral methods for monthly streamflow forecasting in Northwest China, (2) to select the appropriate training period for the models, and (3) to compare the three models and select the best model for streamflow forecasting in Northwest China.

## 2. Methods

Suppose there is a streamflow series, *y*(*t*). Convert it to the frequency domain *f*. If *f* is considered a random variable, the spectral density function is normalized as a probability function. Burg entropy can then be expressed as:(1)$$H_{B}(f)=-\int_{-W}^{W}\ln\left[p(f)\right]df$$
where *W* = 1/2∙Δ*t* is the Nyquist fold-over frequency and Δ*t* is the sampling period.

The definition of configuration entropy is similar to Burg entropy and is defined as:(2)$$H_{C}(f)=-\int_{-W}^{W}p(f)\ln\left[p(f)\right]df$$

With the given prior spectral density function *q*(*f*), the relative entropy can be defined as:(3)$$H_{R}(f)=\int_{-W}^{W}q(f)\ln\left[q(f)/p(f)\right]df$$

The prior spectral density is like background noise in the peak of observed periodicity. When spectral density is uniform, the relative entropy reduces to a configuration entropy.

The processes shown in Figure 1 mainly include (1) calculating parameters; (2) determining spectral density function; (3) extending autocorrelation function; and (4) forecasting streamflow and comparing the three methods for the selection of the most appropriate method.

### 2.1. Deriving Spectral Density Function

In order to obtain the least biased spectral density, under some given constraints, the Burg and configuration entropies are maximized while the relative entropy is minimized before spectral density estimation. According to the relationship of spectral density function and autocorrelation function, the constraints could be given as:(4)$$\rho (n)=\int_{-W}^{W}p(f)e^{i2\pi fn∆t}df,-N\leq n\leq N$$
where $i=\sqrt{-1}$, ρ(n) is the autocorrelation function of *n*-th lag; *N* usually equals 1/4 to 1/2 of the streamflow length series.

Subject to the constraints, entropy can be maximized or minimized using the Lagrangian function, which can be formulated as:(5)$$L(f)=H(f)-\sum_{n=-N}^{N}\lambda_{n}\left[\int_{-W}^{W}p(f)e^{i2\pi fn∆t}df-\rho (n)\right]$$
where *λ_n_* is the Lagrangian multiplier and *H*(*f*) is the entropy function. The partial derivatives of *L* to the spectral density are taken and then equated to zero. The least biased spectral densities obtained by maximizing Burg entropy and configuration entropy and minimizing relative entropy respectively, are expressed as follows:(6)$$p_{B}(f)=-\frac{1}{\sum_{n=-N}^{N}\lambda_{n}\exp(-i2\pi fn∆t)}$$
(7)$$p_{C}(f)=\exp(-1-\sum_{n=-N}^{N}\lambda_{n}e^{i2\pi fn∆t})$$
(8)$$q_{R}(f)=p(f)\exp\left(-1-\sum_{n=-N}^{N}\lambda_{n}e^{i2\pi fn∆t}\right)$$

### 2.2. Calculating Lagrangian Multipliers

The methods of calculating Lagrangian multipliers are different due to the variation in the forms of spectral densities. For Burg entropy, Levinson–Burg algorithms [6,17] are applied to determine Lagrangian multipliers. While in the case of configuration entropy and relative entropy, cepstrum algorithms are applied to calculate Lagrangian multipliers. By taking the inverse Fourier transform of the log magnitude of Equation (8), we can obtain:(9)$$\int_{-W}^{W}\left[1+\ln q\left(f\right)-\ln p\left(f\right)\right]e^{2i\pi fn∆t}df=\int_{-W}^{W}\left(-\sum_{s=-N}^{N}\lambda_{n}e^{2i\pi fs∆t}\right)e^{2i\pi fn∆t}df$$

Take the prior and posterior cepstrum of autocorrelations which are transformed from the prior and posterior spectral densities and expressed as *e_q_(n)* and *e_p_(n)* in the following equations:(10)$$e_{q}(n)=\int_{-W}^{W}\ln q_{R}\left(f\right)e^{i2\pi fn∆t}df$$
(11)$$e_{p}(n)=\int_{-W}^{W}\ln p\left(f\right)e^{i2\pi fn∆t}df$$

Then Equation (9) can be abbreviated as:(12)$$\delta_{n}+e_{q}\left(n\right)-e_{p}\left(n\right)=-\sum_{s=-N}^{N}\lambda_{s}\delta_{s-n}$$
where *δ_n_* is a delta function.

Lagrangian multipliers can be solved using *N* linear functions from Equation (12) of the relative entropy(13)$$\lambda_{n}=\left\{ \begin{array}{l} -1-e_{q}\left(n\right)+e_{p}\left(n\right)\;;n=0\\ -e_{q}\left(n\right)+e_{p}\left(n\right)\;\;\;\;;n\neq 0 \end{array}\right.$$

For configuration entropy, *e_q_* = 0, Lagrangian multipliers can then be calculated by:(14)$$\lambda_{n}=\left\{ \begin{array}{l} -1+e_{p}\left(n\right)\;;n=0\\ e_{p}\left(n\right)\;\;;n\neq 0 \end{array}\right.$$

### 2.3. Forecasting Streamflow

BESA allows autocorrelation to expand as a linear combination of previous autocorrelation parameters with predicted coefficients. $\rho_{N+k}$ can be expressed as:(15)$$\rho_{N+k}=-\sum_{j=1}^{m}a_{j}\rho_{N+k-j}$$
where *a_j_* is obtained using the reflection recursion method proposed by Burg [18].

For the configurational and relative entropies, the autocorrelations are extended as:(16)$$\rho_{N+k}=\sum_{j=1}^{m}\frac{k}{N+k}e(j)\rho_{N+k-j}$$
and
(17)$$\rho_{N+k}=\frac{e_{p}(N+k)}{2}+\sum_{j=1}^{m}\frac{k}{N+k}e_{p}(j)\rho_{N+k-j}$$

According to the extended autocorrelation functions, the forecasting equations of the three spectral entropies methods are obtained as follows:(18)$$y(T+k)=\sum_{j=1}^{m}a_{j}y(T+k-j)$$
(19)$$y(T+k)=\frac{1}{2}\sum_{j=1}^{m}\frac{k}{T+k}e(j)y(T+k-j)$$
(20)$$y\left(T+k\right)=\frac{C_{p}(T+k)}{2}+\sum_{j=1}^{m}\frac{j}{m+1}e_{q}(j)y(T+k-j)$$
where *C_p_*(*T* + *k*) is the cepstrum of streamflow series, and it always equals $\frac{1}{2}e_{p}(N+k)$. *m* is the order of the model, which is determined by *BIC* criteria.
(21)$$BIC(m)=N\ln\sigma_{\varepsilon }{}^{2}+m\ln N$$
where *N* is the length of streamflow series and $\sigma_{\varepsilon }{}^{2}$ is the variance of residual of observed and forecasted streamflow.

### 2.4. Evaluating the Precision of Forecasting Results

In this paper, we selected average relative error (*RE*), root mean square error (*RMSE*), correlation coefficient (*R*) and determination coefficient (*DC*) as evaluation indicators for the forecasted results. The *RE*, *RMSE*, *R* and *DC* are expressed as:(22)$$RE=\frac{1}{N}\sum_{t=1}^{n}\left|\frac{x\left(t\right)-f\left(t\right)}{x\left(t\right)}\right|$$
(23)$$RMSE=\sqrt{\frac{\sum_{t=1}^{n}{\left[x\left(t\right)-f\left(t\right)\right]}^{2}}{n-1}}$$
(24)$$R=\frac{\sum_{t=1}^{n}\left(x\left(t\right)-\bar{x}\right)\left(f\left(t\right)-\bar{f}\right)}{{\left[\sum_{t=1}^{n}{\left(x\left(t\right)-\bar{x}\right)}^{2}\sum_{t=1}^{n}{\left(f\left(t\right)-\bar{f}\right)}^{2}\right]}^{0.5}}$$
(25)$$DC=1-\frac{\sum_{t=1}^{n}{\left|x\left(t\right)-f\left(t\right)\right|}^{2}}{\sum_{t=1}^{n}{\left|x\left(t\right)-\bar{x}\right|}^{2}}$$
where $\bar{x}$ represents the average value of observed streamflow *x*(*t*), $\bar{f}$ represents the average value of the forecasted streamflow *f*(*t*), and *n* is forecasting period (month). According to the “forecasting norm for hydrology intelligence”, the determination coefficient (*DC*) is classified into three levels as shown in Table 1.

## 3. Application

### 3.1. Data Preprocessing

Observed streamflow data from five hydrological stations, Yingluoxia, Zamusi, Jiutiaoling, Xiangtang and Tangnaihai, in Northwest China were selected to verify these three spectral entropy methods. These five hydrological stations are located in the Yellow River, Heihe River and Shiyang River, respectively. Tangnaihai station is located at the mainstream of the Yellow River, while Xiangtang is located at the tributary of the Yellow river. Zamusi and Jiutiaoling stations are situated on the Shiyang River. Yingluoxia station is located at the Heihe River and it marks the boundary between the upstream and middle reaches [1]. Basic information on the five hydrological stations are shown in Figure 2 and Table 2.

The entropy spectral analysis model belongs to the autocorrelation methods, and the input data should be a standardized stationary random sequence. To meet the requirement, the streamflow sequences should be transformed using the Box–Cox method. Box–Cox transformation can eliminate data skewness and make data errors present a normal distribution [17]. In addition, standardized transformation was also performed on the sequences.

To test whether transformed sequences were stable, we verified the unit root of sequences. If the unit root exists in the sequence, it is not a stationary random sequence and vice versa [19]. The adftest function in the econometric toolbox of MATLAB 2010b (2010b, MathWorks, Beingjing, China, https://ww2.mathworks.cn/products/matlab-online.html) was used to test whether the unit root exists. The adftest function assumes that the unit root does not exist in the sequence. If the hypothesis is true, the logical value of *H* is 1 and the confidence can be returned. If the hypothesis is false, the logical value of *H* is 0. The test results of all transformed streamflow sequences for five hydrological stations show that all the sequences are stable and homogeneous (Table 3).

### 3.2. Determining Training Period

In previous research, the training periods were always less than 100 months, and very few papers discussed the influence of the training period on the forecasted results. In this paper, we selected the observed streamflow data from the years 2008 to 2012 as the validation period. Additionally, observed data from 3 to 40 months were selected as a training period to evaluate the influence of the training period on forecasted results. In order to determine the period of models, the relationship between the different training periods and the optimal order of the models are explained in Figure 3. As seen in Figure 3, when the training period is short, the optimal fitting order of models is lower, and then the optimal order of models tends to be stable with the increase of training period.

Beyond that, the relationship between the training period and the *DC* of the validation period were explored (Figure 4). The forecasting effect is weak and not stable enough when the training period is less than 15 years. However, increase in the training period increases and stabilizes the *DC*. In order to make use of the expert opinion to increase the forecasting precision, the calibration period was determined as 26 years.

### 3.3. Estimating Spectral Density

Spectral analysis is a powerful method employed to check the periodicity by finding out the frequency of spectrum peaks. The spectral densities estimated by these three spectral entropy methods were compared to the spectral density estimated by fast Fourier transform (FFT) (Figure 5). Five representative rivers were chosen to show the ability of BESA, CESA, and RESA to estimate the spectral densities. For RESA, a prior spectral density function was hypothesized from data information. The determination process of prior spectral density functions is described in Appendix A. It can be discovered from Figure 5 that all of the stations displayed a peak at frequency 1/12. On the other hand, there were other peaks near frequency 1/4th and 1/6th in the spectral density at Zamusi, Jiutiaoling and Tangnaihai stations.

For uni-peak streamflow series, the BESA, CESA, and RESA can check the periodicity equally as well as FFT. However, for multi-peak streamflow series, the BESA did not perform as effectively in detecting the principal periodicity. On the contrary, the CESA and RESA correctly checked the most significant peak at the 1/12th frequency. However, CESA always neglects all secondary spectral peaks to keep the peak at 1/12th most significant. The RESA detected less significant peaks, and was consistent with the FFT results.

In order to examine whether this variation would affect the forecasting precision, we used these three methods to forecast streamflow in five hydrological stations for selecting the optimal model in northwest China.

### 3.4. Streamflow Forecasting Analysis

Streamflow was forecasted using three spectral entropy methods for five hydrological stations (Figure 6 and Figure 7) with a validation period of five years. The results indicated that the forecasting accuracy was worse in Tangnaihai station where the *DC* is less than 0.6 (Table 4) and belongs to level C compared with the other four stations. The reason for this may be that the catchment area of Tangnaihai station is much wider than other stations. Moreover, the intensive anthropogenic activities might also have a severe impact on the streamflow of Tangnaihai station. Therefore, it is difficult to accurately forecast streamflow with only streamflow from previous months using autoregression-based models.

By comparing the forecasting accuracy of the three models for five hydrological stations during the validation period, we discovered that the rank of forecasting accuracy with the evaluation criteria of *DC*, *RMSE* and *R* for the three models was in the order RESA > CESA > BESA for Yingluoxia station (Table 4). However, for Zamusi station of Shiyang River, the accuracy of the CESA model was higher than the other models (Table 4, Figure 6). For the remaining three hydrological stations, the accuracy was similar for the three models, and the RESA model was more accurate than CESA and BESA models using *DC*, *RMSE* and *R* criteria. However, the *RE* between the observed streamflow and forecasted streamflow using BESA was smaller than other methods. The reason for this is that *RE* reflects the linear error between observed values and forecasted values, while the *RMSE*, *R* and *DC* reflect the quadratic power error between observed values and forecasted values. When the forecasting error of the flood season was smaller, the *RMSE*, *R* and *DC* would be effective. However, when the forecasting error of the non-flood season were smaller, *RE* would be better.

To verify this conjecture, the whole period was divided into flood season from July to October and low-flow season from January to June, November, and December in each year. We extracted the forecasted streamflow of the non-flood season and compared it with the observed streamflow in the five stations (Table 5). As shown, BESA performs better than other methods. During the low flow season, the advantage of BESA over the others was significant, where the streamflow was forecasted close to the observation. However, the overall forecasting accuracy of the RESA model and the CESA model was higher. At the same time, because the streamflow forecasting itself serves as the optimal allocation of water resources, the annual or flood runoff prediction was more meaningful. As a whole, the RESA model can better adapt to the streamflow forecasting for the five hydrological stations in northwest China. Combining precipitation as a predictor, selecting one or more models with high accuracy in the flood season, and using the entropy spectrum model and its combination [1] to forecast streamflow could be a future research direction.

## 4. Conclusions

In this paper, the BESA, CESA, and RESA models were applied for spectral analysis and streamflow forecasting in northwest China using monthly streamflow data from five hydrological stations. The estimated spectral density and prediction accuracy of the three methods were compared based on the optimal length of the training period. The spectral density functions of the BESA, CESA and RESA was smoother than that of FFT, and all of them can clearly estimate the 12 month primary period of monthly streamflow sequence without deviation.

However, the spectral density function of BESA could not detect the other significant secondary periods, while that of CESA could detect the secondary periods for multi-period sequences despite a certain degree of leakage. By comparing these three entropy spectral methods, we discovered that all of these methods could forecast streamflow accurately. Among them, the RESA model has the highest prediction accuracy, followed by the CESA model.

Due to the lack of data, this paper only applied the entropy spectral theory to the monthly streamflow forecasting of few rivers in northwest China. In future research, three entropy spectral analysis methods can further be applied to other rivers to verify the applicability of the three entropy spectral analysis methods in the forecasting of monthly streamflow in China.

## Figures and Tables

**Figure 1 entropy-21-00132-f001:**
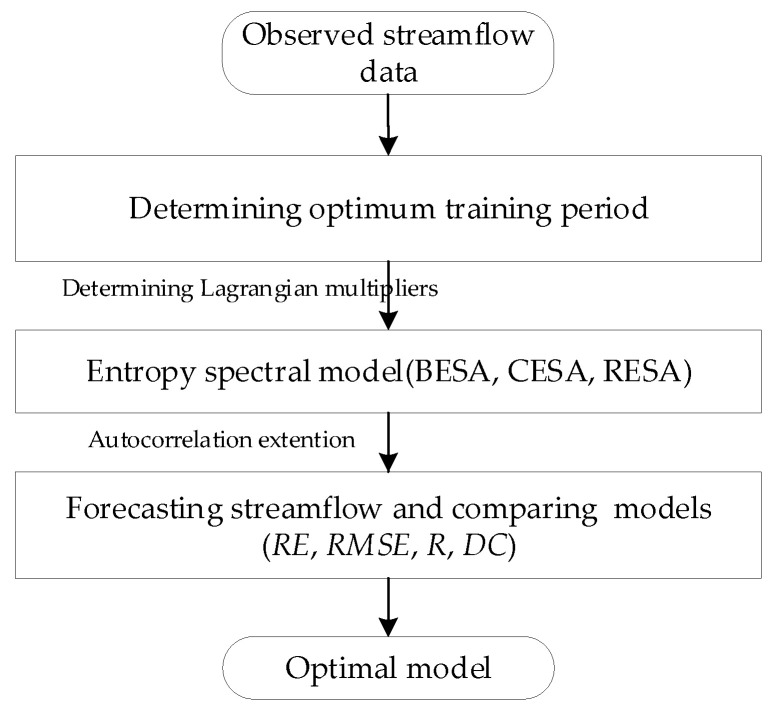
The flow chart of streamflow forecasting using entropy spectral method. *RE*: average relative error; *RMSE*: root mean square error; *R*: correlation coefficient; *DC*: determination coefficient.

**Figure 2 entropy-21-00132-f002:**
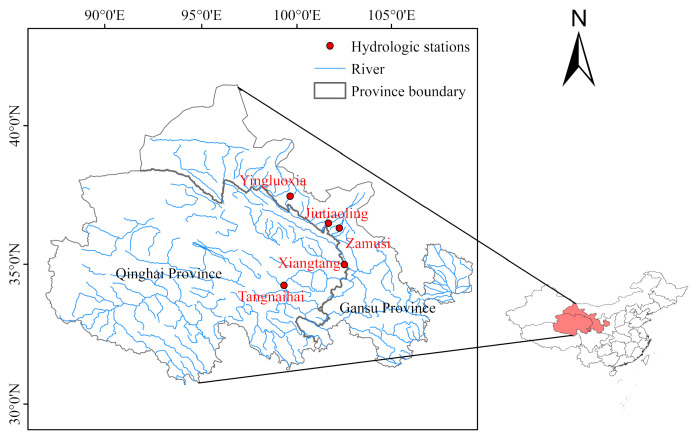
Location of hydrologic stations in Northwest China.

**Figure 3 entropy-21-00132-f003:**
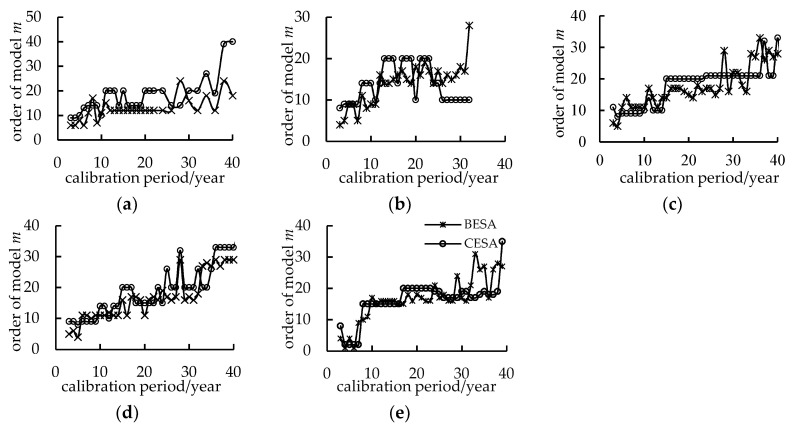
The model order corresponding to the different calibration periods. (**a**) Yingluoxia station; (**b**) Jiutiaoling station; (**c**) Zamusi station; (**d**) Xiangtang station; (**e**) Tangnaihai station.

**Figure 4 entropy-21-00132-f004:**
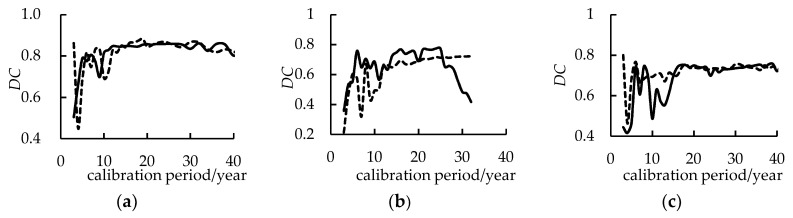
Evaluation index (*DC*) corresponding to different lengths of calibration period. (**a**) Yingluoxia station; (**b**) Jiutiaoling station; (**c**) Zamusi station; (**d**) Xiangtang station; (**e**) Tangnaihai station.

**Figure 5 entropy-21-00132-f005:**
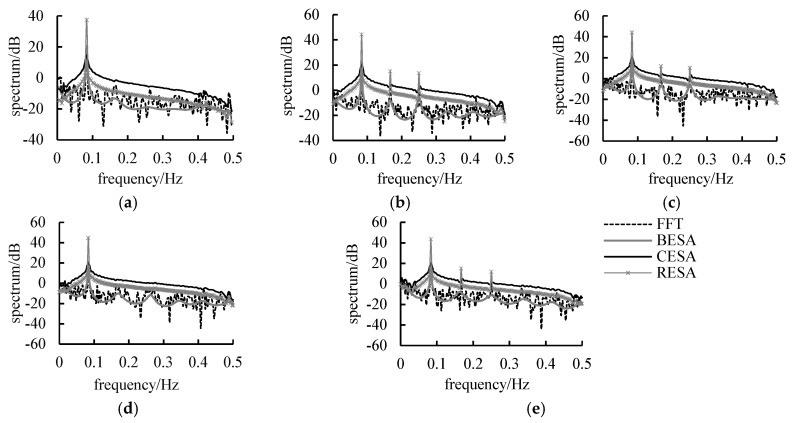
Spectral density estimated by BESA, CESA, RESA and fast Fourier transform(FFT) method for five hydrological stations in Northwest China. (**a**) Yingluoxia station; (**b**) Jiutiaoling station; (**c**) Zamusi station; (**d**) Xiangtang station; (**e**) Tangnaihai station.

**Figure 6 entropy-21-00132-f006:**
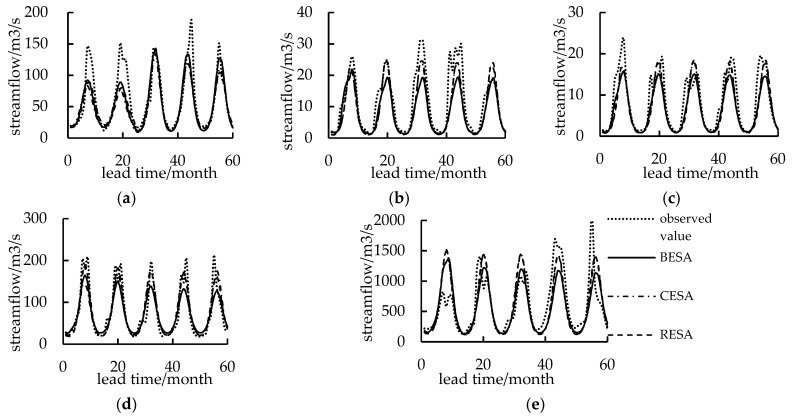
Streamflow forecasting using entropy spectral methods for five hydrological stations. (**a**) Yingluoxia station; (**b**) Jiutiaoling station; (**c**) Zamusi station; (**d**) Xiangtang station; (**e**) Tangnaihai station.

**Figure 7 entropy-21-00132-f007:**
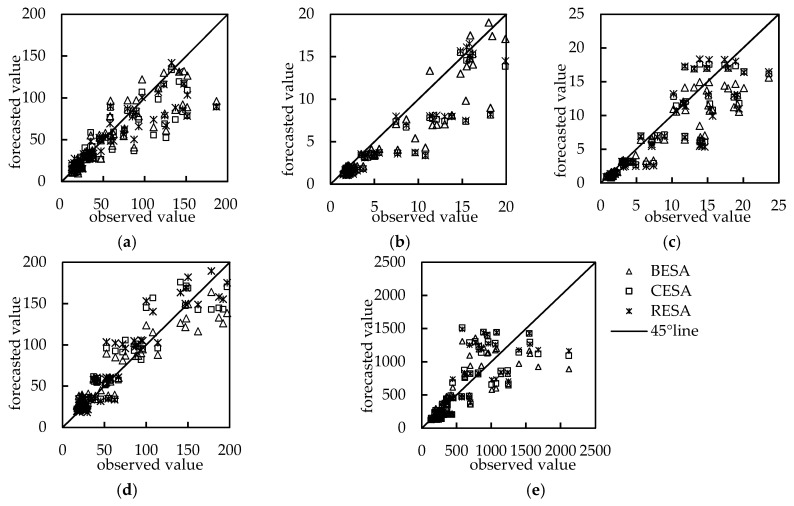
Comparison between observed and forecasted streamflow. (**a**) Yingluoxia station; (**b**) Jiutiaoling station; (**c**) Zamusi station; (**d**) Xiangtang station; (**e**) Tangnaihai station.

**Table 1 entropy-21-00132-t001:** Model forecasting accuracy rating.

Criterion	A	B	C
*DC*	≥0.9	0.9~0.7	0.7~0.5

**Table 2 entropy-21-00132-t002:** Basic information of streamflow data for selected hydrologic stations [1].

Hydrologic Station	Longitude	Latitude	River	Catchment Area (km^2^)	Control Area (km^2^)	Annual Runoff (m^3^/s)
Yingluoxia	100°11′ E	38°48′ N	Hei River	130,000	10.009	51
Zamusi	102°34′ E	37°42′ N	Zamu River	851	851	8
Jiutiaoling	102°03′ E	37°52′ N	Xiying River	1120	1077	10
Xiangtang	102°51′ E	36°22′ N	Datong River	15.133	15,126	88
Tangnaihai	100°09′ E	35°30′ N	Yellow River	752,443	121,972	633

**Table 3 entropy-21-00132-t003:** Adftest test results of monthly streamflow in each hydrologic station.

Hydrologic Stations	Yingluoxia	Zamusi	Jiutiaoling	Xiangtang	Tangnaihai
Returned value	1	1	1	1	1
*p* Value	0.001	0.001	0.001	0.001	0.001
Confidence coefficient (%)	99.9	99.9	99.9	99.9	99.9

**Table 4 entropy-21-00132-t004:** Three models’ performance metrics in each of the selected hydrological station.

Hydrological Station	BESA				CESA				RESA			
	*RE*	*RMSE* m^3^/s	*R*	*DC*	*RE*	*RMSE* m^3^/s	*R*	*DC*	*RE*	*RMSE* m^3^/s	*R*	*DC*
Yingluoxia	0.173	18.348	0.934	0.859	0.194	16.807	0.942	0.882	0.196	16.571	0.944	0.885
Zamusi	0.216	3.395	0.924	0.734	0.259	3.303	0.892	0.748	0.268	3.521	0.876	0.714
Jiutiaoling	0.224	4.972	0.911	0.716	0.273	4.771	0.911	0.739	0.276	4.592	0.911	0.758
Xiangtang	0.260	27.237	0.924	0.797	0.232	25.760	0.907	0.818	0.234	22.636	0.928	0.859
Tangnaihai	0.315	303.303	0.765	0.545	0.324	302.749	0.780	0.547	0.326	291.922	0.796	0.579

**Table 5 entropy-21-00132-t005:** Three models’ performance in non-flowed metrics in each selected hydrological station.

Hydrological Station	BESA				CESA				RESA			
	*RE*	*RMSE* m^3^/s	*R*	*DC*	*RE*	*RMSE* m^3^/s	*R*	*DC*	*RE*	*RMSE* m^3^/s	*R*	*DC*
Yingluoxia	0.179	8.01	0.943	0.870	0.239	10.54	0.939	0.787	0.231	10.51	0.943	0.788
Zamusi	0.224	3.00	0.936	0.726	0.255	3.15	0.901	0.692	0.263	3.47	0.887	0.665
Jiutiaoliing	0.229	4.02	0.863	0.654	0.257	4.42	0.864	0.631	0.268	4.40	0.864	0.636
Xiangang	0.245	12.72	0.857	0.807	0.233	13.11	0.841	0.750	0.259	15.36	0.847	0.683
Tangnaihai	0.261	153.08	0.802	0.630	0.276	165.70	0.813	0.567	0.284	158.23	0.827	0.605

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
