# Peer review of "Application of Entropy Spectral Method for Streamflow Forecasting in Northwest China"

_entropy, 2019, doi:10.3390/e21020132_

Round 1

Reviewer 1 Report

The paper presents results of research on the application of entropy in predicting streamflow of five rivers located in the north-western part of China. Undoubtedly, the topic is interesting and deserves attention, and the Authors have made some efforts to investigate it. However, in my opinion the submitted paper has shortcomings, which make the paper unsuitable for publication in the present form. Major drawbacks are as follows:

1. A map showing the geographical location of the study area in China is required.

2. Please explain if the data sets used in the study were checked for their homogeneity and stationarity, which are considered prerequisites for further hydro-meteorological analyses.

3. Please write clearly what periods were taken for model calibration and validation, respectively. It is very difficult to find that information in the text.

4. In Abstract he Authors introduce “decision coefficient” (DC) (p. 1, l. 17), which is also called by them “determination coefficient” (p. 6, l. 140) and “forecasting accuracy” (p. 6, l. 147). It is very confusing. Do the three different names refer to the same value? On what basis were the DC values in tables 3 and 4 calculated? Please explain.

5. The English language is poor and requires corrections, preferably by an English native-speaker.

In general, it is recommended to accept the submission for publications after major revision.

Author Response

Dear reviewer,

Thank you very much for your recommendation and comments. These comments are very valuable and helpful for revising and improving our paper. A major revision has been made to our manuscript in accordance with these comments. The response comments and the corresponding correction to the paper are explained in detail. We hope our efforts will be met with approval.

The specific responses are in the attached file.

Once again, thank you very much for all your help in reviewing our paper.

Reviewer 2 Report

I have read over this paper quite carefully and  unfortunately there are some major oversights with respect to this work. I have therefore rated this paper as "not ready for publication" at this time. My major issues are that I do not see any references (past the introduction) to what was used as a prior for the RESA, and more importantly, the connection to Chui and Singh's (2016, 2017) work was not really developed. I think these latter authors carried out a much better analysis. Lastly the english grammar and spelling needs a great deal of improvement. I would suggest the authors try to find an english spelling colleague and carry out careful editing  before it  submitted next time

Author Response

Dear reviewer,

Thank you very much for your recommendation and comments. These comments are very valuable and helpful for revising and improving our paper. A major revision has been made to our manuscript in accordance with these comments. The response to comments and the corresponding correction to the paper are explained in detail. We hope our efforts will be met with approval.

 The specific responses are in the attached file.

Once again, thank you very much for all your help in reviewing our paper.

Round 2

Reviewer 1 Report

Corrections made by the Authors are satisfactory. I recommend the paper for publication in the present form. 

Author Response

Dear reviewer

Thank you very much for you suggestions, which are help for our article. 

Yours sincerely,

Gengxi Zhang

Reviewer 2 Report

Improvements to the manuscript have been made. There are a few issues still with a writing that could use a bit of improvement and I'll just leave it to the author's to very carefully go through and edit their manuscript again perhaps the editor staff can also help with that. I found there was some more discussion on the issue related to Prior probabilities in the manuscript I don't have time really to go through all the theoretical implications of doing that unfortunately. Lastly I know that this is mostly a case history and I'm not sure to what extent the journal entropy is willing to entertain those kinds of things I would have liked to have seen much more of a theoretical type of analysis. But I think at this stage will leave it up to the readers and the editorship ship to make that decision on its value. The authors may want to also read over and perhaps reference Woodbury and ulrych,s paper in stochastic hydrology hydrauics 1998.

Author Response

Dear reviewer

Thank you very much for your suggestions, witch are helpful for our paper. We readed the paper you recommended and cited it. It is very helpful for our paper's improvement in introduction. We have carefully check and modified the manuscript. Thanks again for your suggestions.

Yours sincerely,

Gengxi Zhang